# Dear admission committee…: Which moves in application essays predict student master grades?

**Timon de Boer**[1,2]*, **Frank Van Rijnsoever**[1], **Hans de Bresser**[3]

**1** Copernicus Institute of Sustainable Development, Utrecht University, Utrecht, Utrecht State, The Netherlands, **2** Dialogic Innovation and Interaction, Utrecht, Utrecht State, The Netherlands, **3** Earth Sciences Department, Utrecht University, Utrecht, Utrecht State, The Netherlands

* t.h.j.deboer@uu.nl

**Data Availability Statement:** Data cannot be share publicly because of Dutch privacy laws as it contains personal information of individuals. It is not allowed to publicly make data available, as the dataset includes data on grades, dates, ages and

## Abstract

Application essays are a commonly used admission instrument for students entering higher education. The quality of the essay is usually scored, but this score is often subjective and has poor interrater reliability due to the unstructured format of the essays. This results in mixed findings on the validity of application essays as an admission instrument. We propose a more objective method of using application essays, using Latent Dirichlet Allocation (LDA), a text mining method, to distinguish seven moves occurring in application essays written by students who apply to a master degree program. We use the probability that these moves occur in the essay to predict study success in the master. Thereby we answer the following research question: What is the effect of discussing different moves in students' application essays on the student grades in a master program? From the seven different moves (functional unit of text) we distinguished, five of which have a significant effect on student grades. The moves we labeled as 'master specific' and 'interest to learn' have a positive effect on student grades, and the moves we labeled as 'research skills', 'societal impact' and 'city and university' have a negative effect. Our interpretation of this finding is that topics related to intrinsic motivation and specific knowledge, as opposed to generic knowledge, are positively related with study success. We thereby demonstrate that application essays can be a valid predictor of study success. This contributes to justifying their use as admission instruments.

## 1. Introduction

Since, the 1990s, the number of academic studies on the predictive validity of admission criteria for university students has steadily increased [1, 2]. Among these, studies on admission criteria that test the students' cognitive skills, such as grades or standardized tests, are dominant [1, 3–5]. However, academics and practitioners have noted that assessing only the students' cognitive skills yields an incomplete representation of their abilities, and increases the likelihood of incorrect admission decisions [4]. For this reason, studies into non-cognitive

nationalities which can be used to identify subjects. Of course, the data will be available upon request by the first author or the data steward of the faculty. The latter can be reached at a.j.dikker@uu. nl.

**Funding:** The author(s) received no specific funding for this work.

**Competing interests:** The authors have declared that no competing interests exist.

admission criteria have become increasingly prevalent [6, 7]. Examples of such non-cognitive criteria are teamwork, communication, empathy and motivation [8].

Of these con-cognitive criteria, motivation is among the most prominent. A student's motivation is well known to affect the study success and study behavior [9, 10]. A commonly used admission instrument to test this criterion is the students' application essay, also known as personal statement or motivation letter [11]. Universities often require students to write an application essay during the admission process. In the essay students explain different reasons why they should be admitted to the study program. These reasons are considered in the admission decision. For students, application essays provide an opportunity to state why they are suitable for a program. This can be done by writing about their non-cognitive skills, stressing their motivation for entering the program, or highlighting qualities that are not immediately discernable from their prior education and grades, such as personal values, extracurricular activities or work experience [12–14]. Application essays also allow students to provide context to aspects of their application. For example, why they experienced a delay in completing their Bachelor program. Moreover, application essays are well suited for students to reflect on themselves. The ability to self-reflect is often associated with successful students [15, 16].

To admissions offices, application essays are a relatively efficient means to gain insight in why a student wants to study a particular program, to understand their context, and to better assess cognitive and non-cognitive skills. In addition, application essays give admissions offices some insights in the writing skills of the student. Finally, if used properly, letters can give insights in the socio-economic background of the student, which can help to keep the admissions sufficiently diverse [17].

However, some scholars doubt if application essays are a useful admission instrument [18, 19] because the empirical evidence on the use and the predictive validity of application essays for study success is scarce and with ambiguous results [17, 19]. Application essays may also favor students from an advantaged socio-economic background because of their particular writing skills and stronger access to useful advice [20]. Furthermore, the format of an application essay is often relatively unstructured, meaning there is not provided template, leading to sizable differences between them. This unstructured format and the sheer volume of text make it difficult to systematically assess the effect of application essays on study success for admissions offices, and thereby to evaluate the usefulness of application essays as an admission instrument. This criticism indicates a need to further the study predictive validity of the commonly used admission instrument.

Recently, scholars shifted their attention to studying the contents of application essays. This is an important shift, as earlier studies on this topic often measure the quality of application essays with a rating of one or more assessors [19, 21, 22] and sometimes find positive effects on student grades [23, 24]. However, the interrater reliability of such ratings is often poor [21, 22, 25]. There are a number of studies studying the contents of application essays, identifying topics discussed in them [26–28], but do not relate these to grades. Furthermore, these paper rely on more qualitative methods, namely discourse and genre analysis. Using a quantitative approach, Pennebaker et al. [11] study the language students use in application essays using linguistic analysis. They find that students that use categorical language, about complex concepts and objects, acquire higher grades. Lower grades are acquired by students who write more about personal narratives. Alvero et al. [17] also use a linguistic analysis to show that the writing style and contents of the application essay are strongly related to household income and SAT scores. Hoblos and Kakkonen et al. [29, 30] use a LDA to replace assessor rating with an automatic grading process for the quality of the application essay, and predict student success accurately with this method. The paper of Brown [31], which combines qualitative and

quantitative methods, reports that students dedicating more space in their application essay to their research experience later become more successful in their study program.

However, to our knowledge, there is no research so far that directly links the topics occurring in an application essay to subsequent academic success in a master program. For example, we do not know if discussing expectations and ambitions for the master's program is more likely to lead to study success than discussing professional development after the master's program, while both are likely topics in an application essay. Examining the content of application essays, and relating these to study success can give admission officers an heuristic to make better use of this admission instrument.

To fill this knowledge gap, we analyze the application essays from four cohorts of in total 820 graduate students from a large university in the middle of the Netherlands. We do so by using Latent Dirichlet Allocation (LDA). This text mining application seeks patterns within the texts by looking at the co-occurrence of words. This enables us to identify topics that are discussed in the letters, and to assign probabilities to topics in letters. We then link these topic probabilities to grades scored in the master as indicator for study success. This allows us to formulate an answer to the following research question:

**What is the effect of discussing different moves in students' application essays on the student grades in a master program?**

This paper contributes to student admission literature by filling knowledge gaps on one of the most commonly used, but most uncommonly studied, admission criteria: application essays. Our LDA approach ends up yielding seven moves, which are in our case, units of text with the purpose to gain admission to a master program [26]. of which five are significantly related to student grades. We report that two moves related to intrinsic motivation, which we labeled 'master specific' and 'interest to learn', have a positive effect on student grades. Three moves we labeled as 'research skills', 'societal impact' and 'city and university' have a negative effect on student grades. This confirms existing theory that intrinsic motivation is an important driver of study success. Our findings have practical implications for universities, and influences their use of application essays as a admission criterion. First, we find support that application essays can indeed be used as a valid admission instrument. Second, we report a number of topics that influence student grades, which can thus be used as evaluation criteria during the admission process. Third, we argue that intrinsic motivation and discussing specific knowledge, as opposed to generic knowledge, are important overarching admission criteria, which should be considered in the admission decision. Application essays can be a useful instrument to evaluate both.

## 2. Theoretical background

This chapter provides theoretical context for the variables used in this study. We first define our dependent variable: student success. We then provide theoretical background for our independent variables: the themes students write about in their essay. We draw from existing studies on this topic, to gain insight in the themes students discuss in their essays. This way, we can later observe if the topic we distill from essays using text mining are similar to the topics already discussed in academic literature. We finally discuss why we included the control variables we used in this study.

### 2.1. Defining student success

We define student success as 'academic achievement, engagement in educationally purposeful activities, satisfaction, acquisition of desired knowledge, skills, and competencies, persistence,

and attainment of educational objectives [32, p. 7]'. From this definition it follows that it needs to be assessed if these educational objectives are achieved. Most educational objectives are commonly achieved and assessed by completing a series of educational activities, such as courses and a thesis, which culminate in a degree. Therefore, degree attainment is a common way to measure student success. After all, students go to university to acquire knowledge and skills in a coherent program, and need to complete the entire program successfully to fully master the contents of the program. However, simply attaining a degree by finishing courses does not fully capture how proficient students' are at acquiring knowledge. Many universities and students' future employers value proficiency as high measure of success. Just because two students both graduate from the same program, does not mean they've reached a similar level of mastery. Grades are a more suitable method of demonstrating mastery and proficiency of desired learning objectives [33]. They are the most commonly used indicator for student success [1], as they are the primary method of indicating how well the student performed in tests and coursework. Grades also reflect students' self-regulatory competencies [34]. Grades are awarded to students throughout the program, which means they provide a more refined insight into the students development over time. In this paper, we will consider attained grades during the master as indicator for student success.

## 2.2. Themes from application essays

Students write admission essays to demonstrate the fit between themselves and their study program of choice in order to gain admission [26]. Prior studies showed that students can cover several themes in their essay, such as their motivation and reasoning for applying to the program and university of their choice, why they are qualified to follow the program, their personality and relevant life experience, and future career goals [26, 28]. Our paper starts out abductively. We discuss four possible themes in the next paragraphs, but we won't use these four themes to code and label the topics we find using our text mining methodology. Moreover, we won't use them to formulate hypotheses. Nonetheless, this chapter can help us to gain a better grasp on what themes application essays contain, and some minor expectations about how these relate to study success. We will observe how these four themes relate to the topics we find using our text mining approach.

**2.2.1. Motivation for applying.** First, students can be intrinsically motivated to study the program. Intrinsic motivation comes from humans' natural inclination towards exploration, mastery and curiosity [35]. Deci and Ryan state that intrinsic motivation arises when tasks are perceived as challenging, leave room for your own interpretation and carried out in relation to people the individual is sympathetic to. These requirements are present in the case of applying to a graduate program, which is why we expect students to discuss their intrinsic motivation in their application essay. Several studies have demonstrated that learning form intrinsic motivation leads to better study results [36–40], so we would expect students who a priori demonstrate their intrinsic motivation to have higher study success.

Motivation can also come from extrinsic sources. Extrinsic motivation is described as doing something because it leads to a separable outcome [35]. In other words, an activity is undertaken *in order to* achieve a goal which is not inherently tied to the activity [41]. For example: a child does the dishes in order to receive an allowance, not because doing dishes is an inherently rewarding and stimulating activity. One extrinsic reason for students to apply to a master's program is to develop social connections and networks [38]. Because of this, the university lifestyle is attractive for students [42]. During their studies, students seek to engage in meaningful discussions with peers. Through this, many students look for and find a sense of belonging in the academic community [42]. Students are also known to include reasons for

choosing a specific university within a specific country, often using phrases like 'this institute is among the best in the world' [43]. We expect students to cover these sources of extrinsic motivation in their essay, and we expect these topics to have a positive relationship with study success.

**2.2.2. Qualifications for studying.** Qualifications for studying involves describing the students' relevant academic and professional preparedness for the study program [26]. This topic is often considered the most important aspect of the application essay by both students and selectors [26, 27]. Applicants need to convince the reader that they possess the right qualifications for the study program they apply for, and to establish an identity as a competent academic [26, 44]. They will do this by listing their academic achievements so far, such as attained degrees, courses and grades or by highlighting specific projects and research experiences related to their field [26, 28]. Students' qualifications are self-reported, which means this topic contains a risk of self-aggrandizement [44]. Despite this, Brown [31] reports that applicants that dedicate more space in their letter to describing their research experience and interests become more successful.

**2.2.3. Future goals.** Future goals refer to the applicant's intended career goals and desired personal development after graduation. This topic differs motivation for applying (section 2.2.1.) in the sense that students describe here *what* their future career goals are instead of arguing that studying will help them realize their career goals. Students will sometimes write about a specific profession they wish to fulfill in the future (for example: Doctor) and what the universities' post-graduation employment prospects are [27, 43, 45]. Attaining a master's degree is known to positively influences students' career trajectory and their financial prospects [39] and students therefore expect an academic degree to speed up career development and lead to faster promotions [46]. It is likely that such reasons are discussed in students' application essays.

In addition, students can also cover other future goals, such as societal contributions they wish to make (for example: combat climate change), or personal growth they want to achieve (for example: becoming a team worker) [27, 28]. Writing about future career plans, societal contributions, and goals for personal growth, shows that students are able to reflect upon themselves and their learning goals. Students with this ability are known to be more successful, which is why we could expect students who write more about this topic to have higher study success [16].

**2.2.4. Personality and life experience.** With this topic, students explicitly describe their unique personality and life experience to distinguish themselves from the large pool of applicants [26]. Students can also write about their personal history, both related or unrelated to the study program [28]. For instance, students with a relative or close friend who followed the study program or has a career in the field will often mention this in their essay [28]. Another topic which can be covered is the students' experience and affinity with teamwork [26]. Students also often discuss their extracurricular activities, (voluntary) work experience and community involvement that offers insight in their abilities and skills related to their field of study [26, 28, 31]. Little is known about how students' descriptions of their personality relate to study success. However, empirical work shows some personality traits, especially conscientiousness, are positively related to study success [47–49]. However, personal narratives and experiences which students describe in their application essays do not link strongly to their conscientiousness [26, 28].

## 2.3. Control variables

We also include six control variables that can, according to the existing literature, have an effect on our dependent and independent variables.

First, we controlled for the length of the application essay because literature shows that longer application essays tend to be more successful [26, 50]. Second, we controlled for the average bachelor grade of the student. We control for bachelor grades because they have a consistently positive effect on student success [5, 18, 51, 52]. This is because, independent of the content of the bachelor's program, grades reflect the self-regulatory competencies that are needed for a student to successfully navigate a bachelor's program [34]. Grades also reflect the students' overall cognitive ability and intelligence [5]. Because grades reflect students cognitive and self-reflective ability, grades are also likely to influence the quality of the application essay. Our third and fourth control variables are the demographic variables age and sex. Age and sex have a well-reported influence on student success, with multiple studies finding that older students and female students perform better [53–56].

Fifth, we controlled for the students' international background. There is a lot of evidence that international students are at more risk of dropping out compared to their non-international peers [57]. This increased drop-out risk can be attributed to language differences and cultural barriers international students might have to overcome, [57]. Financial pressures and risks are also a common reason for study problems for international students [57]. These pressures can be caused by a precarious personal socio-economic position, coupled with high tuition rates compared to the students country of origin. Another factor is the sometimes unreliable financial support by the student's country of origin [57]. All of these problems do not have the same intensity for every international student, as students from the Global South are more at risk than their Northern counterparts. Because of this, we included variables about the students' international background as a control variable.

Finally, we controlled for the students choice of graduate program. Although the graduate programs in our sample are all from the same faculty, there are clear differences between them. It is possible that these different programs attract different types of students who require different types of skills. The topics from these students' application essays might therefore also differ which makes this a suitable control variable.

## 3. Methods

To answer our research question, we identified topics in the application essays of master students using a Latent Dirichlet Allocation. We then related the probabilities that these topics occur in a letter to the grades achieved in the master. This paper therefore has an abductive research design.

### 3.1. Data description and collection

We collected data about application essays, acquired grades and other information from students at the Faculty of Geosciences at Utrecht University in the Netherlands. This multidisciplinary faculty offers research, impact and education "with regard to our Earth system and sustainability, from the local to the global scale" [58]). The faculty offers master programs via three education institutes that respectively focus on earth sciences, sustainable development and human geography. The master programs vary in their emphasis on the natural or social sciences, but all fit under the thematic umbrella of the faculty. To keep the master programs comparable, we look at the eleven two-year master programs of the faculty that are offered in English. These were at the time: [1] Earth Structure and Dynamics, [2] Earth Surface and Water, [3] Earth, Life and Climate; [4] Energy Science, [5] Geographical Information Systems, [6] Innovation Sciences, [7] Marine Sciences, [8] Sustainable Business and Innovation, [9] Sustainable Development, [10] Urban and Economic Geography, and [11] Water Science and Management. Even though every master's program in the sample is unique in its content, their

overall structure is similar. All programs fit in most of their coursework in the first year, whereas the second year consists mostly of internships and a thesis. This further contributes to the comparability of the programs.

The data is from four cohorts of master students, which started studying in 2014, 2015, 2016 and 2017. Data collection was approved by the faculty's ethics committee, who also waived the need for obtaining consent from students. After data cleaning, the sample consists of 2701 students from all over the world of which 2051 graduated and 650 dropped out. To be admitted, students require a degree in a university bachelor program or a tailor-made premaster program that confirms that they have acquired the knowledge, insight and skills at the university Bachelor's level with regards to the Master's program. Students provide the admission office with a list of attained grades, their resume and a personal application essay. There is no minimum required grade, but students with an average grade of a seven or higher are usually admitted. For students with a lower average grade, the decision often comes down to a careful examination of the application essay. For the application essays, no specific requirements (including word count) were formulated. The only suggestion provided to the applicants was to 'not repeat information from the students resume'. All admissions are made on a case-by-case basis. Despite our efforts, we were not able to fully complete the data for each student. This was mostly due to the fact that it was difficult to obtain an reliable average bachelor grade for international students. The number of students that entered the program and for which we're able to reconstruct a complete admission file was 1307. The number of students that finished their master and for which all data was available is 820. These students were divided over the eleven separate master programs.

## 3.2. Measurement

**3.2.1. Dependent variable: Student success.**   We measured study success by looking at the students' attained grades during the program. Grades are obtained throughout the entire educational program and can therefore be measured in various ways, such as the grade for the first or second semester, the entire first year, the last year, the finishing thesis, or most commonly, a student's average grade over the entire program (For example, see [59–61]. The choice of when to measure student grades matters because studies show that admission instruments such as bachelor's program grades often predict grades in the first year of a program very well, but lose validity in the later years of a program [62, 63]. Students progress at different speeds throughout their program and often need to adjust to a new master's program and institute [64–66]. Therefore, it is important to measure grades at various points in the study program. As an indicator for grades, we used the average grade of the student for the entire master's program and their average grades of the first year of the master's program.

**3.2.2. Independent variables: From essays to moves.**   As independent variables, this study uses moves occurring within students' application essays. A move is a functional unit of text, which has an identifiable purpose [26]. In our case, the purpose being to gain admission to a master program. To identify them, we applied Latent Dirichlet Allocation (LDA), which is a generative probabilistic model of a corpus. A corpus consists of a number of documents, which in turn consists of a number of terms. The LDA treats every application essay as a bag of terms where every term has its' own probability of appearing in any of the topics. The LDA can therefore be used to distill topics from a corpus and identify the concepts that run through it [67]. LDA has been used earlier in the context of higher education. For example, Ogihare & Ren [68] used LDA to distill linguistic features from application essays to predict student retention. There are also several scholars that used LDA to automatically grade the quality of application essays [29, 30, 69].

To identify topics, we first prepared the application essays for analysis. Every term was stemmed and lemmatized with the porter algorithm [70, 71], and then transformed into lower case. Furthermore, all numbers and stop words were removed with the using SMART stop-word list from the "tm" R-package. Second, we determined the optimal number of topics following the methods of Arun [72], Griffiths [73] and Cao [74] using the R package 'ldatuning'. This is an important step as an LDA model with an insufficient number of topics is too coarse to identify accurate classifiers. On the other hand, an excessive number of topics could result in a model that is too complex, making interpretation difficult [75].

With this package, we can set a range for a number of topics, and plot the topic density for this range. The topic density indicates the specificity of terms to certain topics. The higher the topic density, the more specific terms are to that topic. Higher topic densities are preferable, because this makes the topics more easy to distinguish from each other. Fig 1 displays the results of this plot. In the bottom graph of Fig 1 a higher number on the y-axis represent a higher topic density. In the top graph the topic density is inverted, so a higher density leads to a lower number on the y-axis. We chose 30 as the final number of topics as this ensures a high topic density while maintaining a manageable number of topics for labeling. We rendered LDAs with several numbers of topics to check the face validity of the quality of the topics. We did so because we wanted to ensure that every individual master program had its' own topic. Seeing as there are eleven master programs in the dataset, we required more topics than usually seen in topic modeling papers [76–78]. The topic density could be marginally improved by creating more topics, but the higher the number of topics, the smaller the gain in density, so there would be relatively little advantage to doing this. In fact, there are occasions where an increase in the number of topics decreases the topic density, especially when the number of topics already exceeds 30. We then used a Gibbs sampling algorithm to run the LDA [79].

For every topic, the most common terms within the topic were extracted and used to label what the topic was about. Every co-author labeled the topics individually, after which the labels were compared. In the rare case that the co-authors disagreed about the label of a topic, a second round of labeling was done to increase the interrater reliability. Topics with highly similar or identical labels were then merged into moves. Merging the topics into moves was done separately by the first and second author. All three authors then checked the final labeling of the moves, after which definitive labels were chosen. We ended up distinguishing seven separate

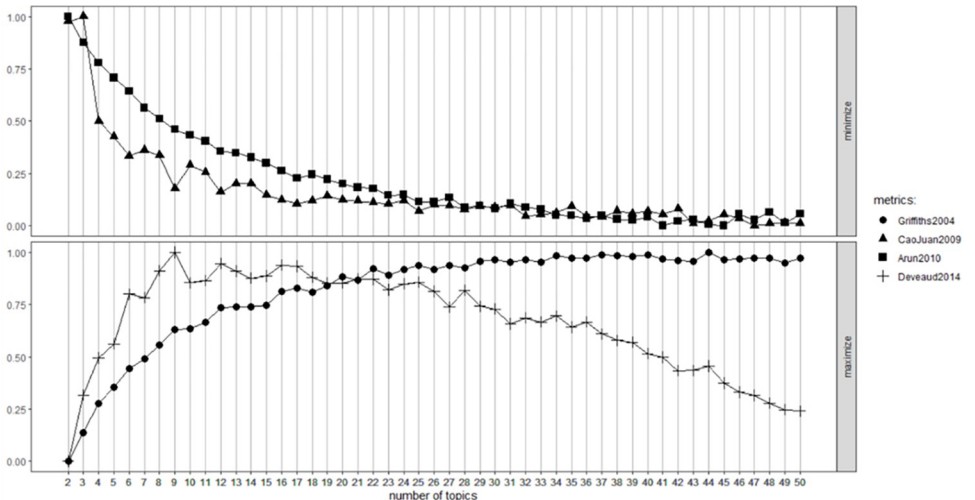

**Fig 1. Plots of optimal number of topics.**

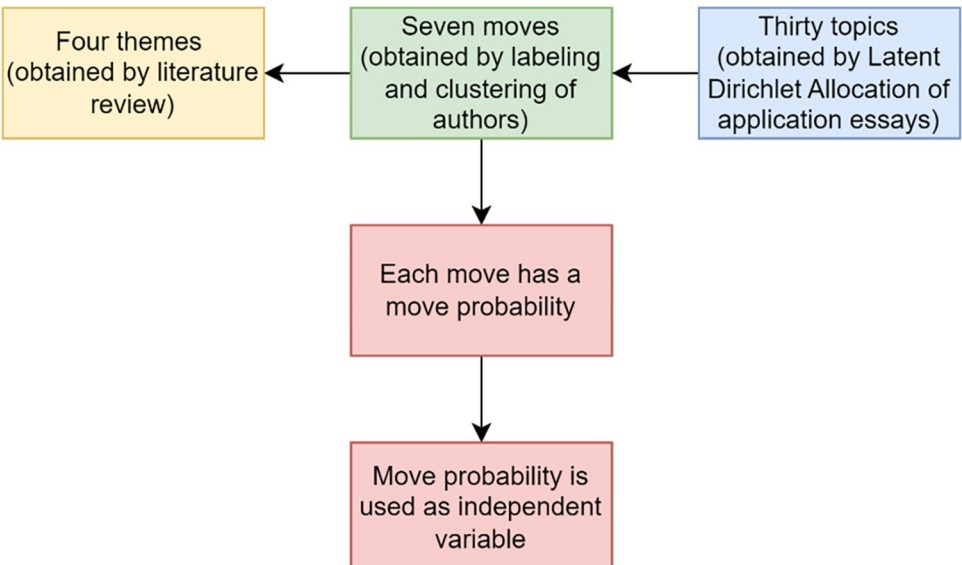

**Fig 2. Schematic overview of topics, moves and themes and how moves are used as independent variable.**

moves from the thirty topics. We discuss these moves, and link them to the themes found in chapter 2 in section 4.1. We schematically show this in Fig 2 below.

For each application essay, the topic probability of every topic and move was calculated. The topic probability is a number between 0 and 1. If the topic probability is 1, it means that all words from the letter belong to that topic. If the topic probability is 0, it means that no words from the letter are from that topic. The probability of a move was calculated by adding up the probabilities of its' underlying topics. These move probabilities were used as predictors of student success during the analysis.

**3.2.3. Control variables.** The average bachelor grades of students who completed a Dutch bachelor's program were extracted from the faculties' data storage application. For the 843 international students the application data were stored in PDF files that listed the full academic transcript of each student. A research assistant manually examined every file and extracted the average grades in the bachelor's program. Information from the Dutch organization Nuffic, which provides information on educational systems across the world, was used to transform international grades into a Dutch equivalent between 1 and 10. The length of the application essay was measured by counting the number of words.

We categorized the students' international using the United Nations geoscheme [80], which classifies countries into regional and subregional groups. The advantage of this classification is that cultural aspect of entire regions are controlled for, without having to rely on very small sample sizes for some countries in our dataset. Unfortunately, for some of these regional and subregional groups the sample size was still very small (for several groups, N was < 5). We therefore manually added together these groups into larger regional or continental groups. In the regressions, we used the category 'Europe' as reference category. The age and sex of each student were known to the faculty as part of the admission process. We refer to a students' biological sex as this is what they filled in their application form. Students have the option to not fill is this in (as one student did) in case they do want to for any reason, like their biological sex is not binary, or their biological sex does not align with their gender identity. Sex was measured as a binary variable, with male students coded as 1 and female as 2. The age of the students was measured by their age in years at the starting day of the master's program. The

**Table 1. Descriptive statistics of variables used in this study.**

| Variable | Mean | SD |
|---|---|---|
| Average master grade | 7.51 | 0.52 |
| Average master grade year 1 | 7.47 | 0.53 |
| Master specific | 0.16 | 0.08 |
| Research skills | 0.02 | 0.02 |
| Prior education | 0.17 | 0.04 |
| Societal impact | 0.09 | 0.04 |
| Interest to learn | 0.10 | 0.03 |
| City & University | 0.07 | 0.03 |
| Extracurricular | 0.03 | 0.02 |
| Word count application essay | 156 | 205 |
| Average bachelor grade | 7.17 | 0.66 |
| Sex (1 = male, 2 = female) | 1.44 | 0.49 |
| Age | 23.8 | 3.44 |
| International background | Europe = 2357<br>Africa = 19<br>Asia = 167<br>Central America = 30<br>Middle-East = 18<br>North America = 53<br>Oceania = 4<br>South America = 53 | 1.33 |
| Master program | Earth, life and climate = 150<br>Energy science = 230<br>Earth, structure and dynamics = 279<br>Earth, surface and water = 289<br>Graphical Information Management and Applications = 231<br>Water science and management = 189<br>Human geography and planning = 97<br>Marine sciences = 140<br>Innovation sciences = 158<br>Sustainable business and innovation = 319<br>Sustainable development = 618 | 3.93 |

choice of master program is extracted from the students' application file. Tables 1 and 2 present the descriptive statistics and correlation matrix of our variables.

## 3.3. Analysis

In the first step of the analyses, we estimated seven beta regression models with the goal of testing the influence our control variables have on the move probabilities. Beta regressions are commonly used when the dependent variable can take values between 0 and 1 [81], as is also the case here. We ran seven beta regressions, with our seven moves from application essays as the dependent variable and all control variables as independent variables (for clarity's sake, we will continue to refer to these as control variables in text). To do this, we first scaled the move probabilities to a number between 0 and 1.

Next, we estimated our main models to predict student grades in their master program. For all models we estimated an ordinary least squares (OLS) multiple linear regression, as the dependent variables tested in these models are continuous. We ran four models, two for each of our dependent variables. The first model for each dependent variable only includes the control variables (models 1 and 3) while models 2 and 4 add the independent variable as main effect.

**Table 2. Correlation matrix of variables used in this study.**

| | First year master grade | Master specific | Research skills | Prior education | Societal impact | Interest to learn | City & University | Extracurr | Word count application essay | Average bachelor grade | Sex (1 = male, 2 = female) | Age | International background |
|---|---|---|---|---|---|---|---|---|---|---|---|---|---|
| **Average master grade** | 0.853 | 0.015 | -0.094 | 0.059 | -0.046 | 0.005 | -0.060 | 0.029 | 0.039 | 0.440 | 0.141 | -0.055 | -0.085 |
| **First year master grade** | | -0.015 | -0.086 | 0.076 | -0.001 | 0.025 | -0.074 | 0.030 | 0.059 | 0.442 | 0.135 | -0.047 | -0.062 |
| **Master specific** | | | -0.182 | 0.064 | -0.132 | 0.002 | 0.032 | 0.088 | 0.362 | -0.013 | -0.032 | -0.036 | -0.078 |
| **Research skills** | | | | -0.149 | 0.049 | -0.059 | 0.070 | -4.254575e-02 | -0.122 | 0.123 | 0.004 | 1.984715e-02 | 0.141 |
| **Prior education** | | | | | -0.203 | -0.013 | 0.035 | 0.040 | 0.149 | -0.014 | 0.004 | -0.048 | -0.091 |
| **Societal impact** | | | | | | 0.049 | -0.045 | 0.044 | 0.171 | 0.121 | 0.078 | 0.018 | 0.241 |
| **Interest to learn** | | | | | | | -0.066 | 0.050 | 0.124 | -0.131 | 0.045 | -0.015 | -0.059 |
| **City & University** | | | | | | | | -0.033 | 0.117 | 0.097 | 0.028 | 0.029 | 0.101 |
| **Extracurricular** | | | | | | | | | 0.043 | -0.034 | 0.047 | -0.030 | -0.028 |
| **Word count application essay** | | | | | | | | | | 0.112 | 0.051 | 0.047 | 0.048 |
| **Average bachelor grade** | | | | | | | | | | | 0.211 | -0.054 | 0.198 |
| **Sex (1 = male, 2 = female)** | | | | | | | | | | | | -0.070 | 0.080 |
| **Age** | | | | | | | | | | | | | 0.209 |

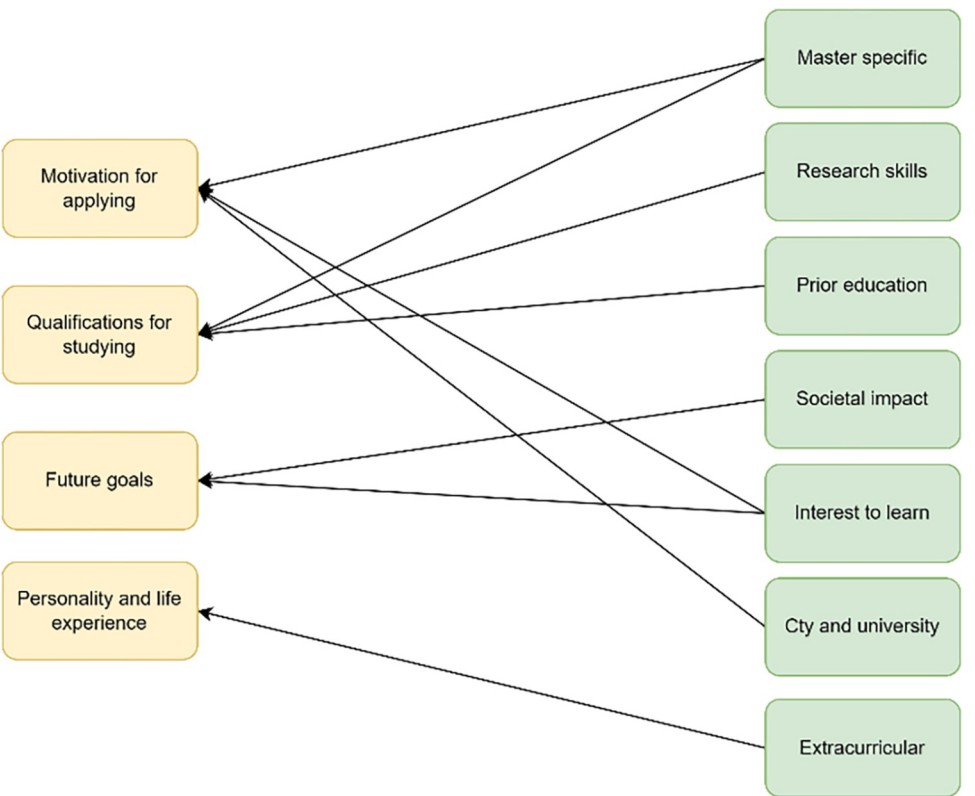

**Fig 3. Connections between themes derived from literature and moves derived from topic modeling.**

## 4. Results

### 4.1. Moves derived from topic modeling

The seven moves are shown on the right side of Fig 3. On the left side of the same figure are the four themes we distilled from literature (section 2.2). We will now discuss the contents of the seven moves we distilled from our topic model, and how we linked these to the themes from existing literature. The linkages between them can be seen in Fig 3. All the moves we derived from our topic model are connected to the themes from literature. Some moves are connected to multiple themes we derived from our literature review, but none are connected to no theme from the literature.

Table 3 presents the results from our beta regressions. Each column presents the estimates of a control variable predicting the probability that a specific move is discussed. Based on the pseudo-$R^2$, we conclude that the model fit varies quite a bit across the seven models. The pseudo-$R^2$ is highest (0.514) for model 1, which measures 'master specific' indicating that the control variables explain the variance of the dependent variables very well. For models 2 and 4 the pseudo-$R^2$ is also higher than 0.2, indicating a good model fit. For models 3, 5, 6 and 7 the pseudo-$R^2$ is below that threshold, implying that the independent variables included do not explain much of the move. We discuss each move, and it's predicting variables. We refrain from discussing explanations for the relationships we find, because such explanations would mostly be speculative.

- The first move we distilled is called 'master specific'. In this move, we found words that describe the contents of the master program. As this move contains the topic probabilities for every individual master program, we multiplied this value with the master program

Table 3. Beta regressions predicting move probabilities with control variables.

| Move probabilities | Master specific [1] | Research skills [2] | Prior education [3] | Societal Impact [4] | Interest to learn [5] | City & University [6] | Extracurricular [7] |
|---|---|---|---|---|---|---|---|
| **Word count application essay** | -0.0004*** | -0.0004** | -0.001*** | 0.0003* | 0.0001 | -0.001*** | -0.0003 |
| **Average bachelor grade** | 0.038 | 0.106** | -0.005 | 0.053 | -0.173*** | 0.055 | -0.057 |
| **Sex (M = 1, F = 2)** | 0.050 | -0.146** | 0.041 | 0.026 | 0.079 | 0.128** | 0.082 |
| **Age** | 0.009 | -0.006 | -0.002 | 0.042*** | -0.005 | 0.028** | -0.003 |
| **International background: Africa** | -0.206 | 0.093 | -0.441 | 0.118 | -0.215 | 0.891*** | 0.076 |
| **International background: Asia** | -0.168* | 1.490*** | -0.196** | 0.320*** | -0.156 | 0.780*** | -0.004 |
| **International background: Central America** | 0.218 | -0.432 | -0.392 | 0.761*** | -0.007 | 0.217 | 0.213 |
| **International background: Middle-East** | -0.372 | 0.465 | -0.266 | 0.447* | -0.077 | 0.269 | -0.360 |
| **International background: North America** | 0.113 | 0.410** | -0.349** | 1.217*** | -0.168 | 0.397** | -0.309 |
| **International background: South America** | -0.117 | 0.020 | -0.214 | 0.667*** | -0.373** | 0.375** | -0.223 |
| **Master program** | Yes | Yes | Yes | Yes | Yes | Yes | Yes |
| **Constant** | -2.812*** | -2.984*** | -0.400 | -3.576*** | -0.148 | -2.647*** | -2.368*** |
| **Observations** | 823 | 823 | 823 | 823 | 823 | 823 | 823 |
| **Pseudo-R$^2$** | 0.514 | 0.208 | 0.081 | 0.317 | 0.072 | 0.120 | 0.085 |
| **Log Likelihood** | 617.697 | 1,400.619 | 387.771 | 836.172 | 524.043 | 507.487 | 1,284.320 |

*p
**p
***p<0.01

dummy variable. In that way, only the topic probability for the 'master specific' topic of the student' master program of choice is used in the regression models. We connected this move to the 'motivation for applying' theme and the 'qualifications' theme because it captures both. After all, students wrote about master specific term to motivate how passionate they are for content of the master, but also to highlight relevant knowledge and experience they already possess on this. The first control (i.e. independent) variable that has a significant effect on this move is the word count of the application essay. This effect is negative, which means that students that have a longer essay write less about 'master specific' content. Second, students from Asia also have a negative significant effect which means they write less about this move compared to students from Europe.

- The second move is called 'research skills'. This move consists of terms that describe the students research and methodological skills. Some of the most common terms in this move are 'research', 'design' and 'method'. We connected this move to the theme 'qualifications',

because it described a skill that qualifies students for entering the master. The control variables word count of the application essay and sex have a negative significant effect on this move, meaning that students with a longer essays and female students write less about their research skills. Students from Asia and North America as well as students with a higher bachelor grade also write significantly more about their research skills.

- The third move is called 'prior education'. With this move, students described which education they have completed prior to applying. This can be about the student's bachelor program, but also about educational institutes the student attended before that. Some of the most common terms in this move are 'school, 'bachelor and 'studi'. We connected this move to the theme 'qualifications', because relevant prior education is an important qualification students highlight when they want to enter a master. There are three control variables that significantly negatively affect the probability of this move occurring: word count of the application essay, and having an international background in Asia or North America'. This means that students from these two continents, and students with a longer essay write less about their prior education.

- The fourth move is called 'societal impact'. This move consists of terms that describe the students desire to create societal impact during and after their master program. They wrote that they want to contribute to solving societal challenges and cover specific societal issues they want to contribute to, such as nature conservation or aiding developing countries. Some of the most common terms in this move are 'develop', 'world', 'problem' and 'live'. We connected this move to the theme 'future goals, because the societal impact students want to make during and after their master lies in the future. Control variables word count and age have a significant positive effect on this move occurring, showing that longer essays and essays written by older students contain more terms about 'societal impact'. Furthermore, every international background has a more significant positive relationship compared with our reference category, Europe, except "Africa. This shows that students from Africa and Europe write less about Societal Impact than all other students.

- The fifth move is called 'interest to learn'. This is a move that described the students motivation for applying and studying. This mostly concerned their proclivity towards acquiring knowledge and learning but also to the attainment of degrees. Some of the most common terms in this move are 'degr', 'work', 'learning' and 'knowledge'. We connected this move to the theme 'motivation for applying', because the desire for students to gain knowledge and learn is a strong motivation for studying. This move also connects to 'future goals' because the desired acquisition of knowledge is often a prerequisite for fulfilling the students' goals in later life. We find significant negative effects for control variables average bachelor grade and 'international background: South America'. This shows that students with a higher bachelor grade and students from South America write less about their interest in further learning.

- The sixth move is called 'city and university'. In this move, students wrote about the city the university lies in and the specific university in this city students' applied to. They commonly write about how both city and the university are well-established and good places for studying. Students used this move to motivate their choice for this specific city and university to study. Some of the most common terms in this move are 'univ, 'name of university city' and 'studi'. We connected this move to the theme 'motivation for applying' because students use it to explain to motivation for studying in this specific city and at this particular university. Table 3 shows that the word count of the application essay has a significant negative effect, meaning that longer essays contain less terms about the city and university the student applied to. There are several other control variables with a significant effect such as sex, age and having international

backgrounds in Africa, Asia, North America and South America'. Students from these backgrounds, female students and older students wrote more about this move.

- The seventh move is called 'extracurricular'. Students used it to highlight their extracurricular activities. The list of activities is quite broad, and includes student associations and boards, voluntary organizations and sports teams. Some of the most common terms in this move are 'activ', 'associ' and 'team'. This move relates to the theme 'personality and life experience', because it captured personal experiences from the students life that take place outside a classroom. None of control variables included in our model influence the probability of this move occurring.

## 4.2. Predicting study success

Table 4 displays the results for our OLS models that predict study success. For all models the adjusted $R^2$ was higher than 0.2, indicating good model fit. Moreover, the moves add a

Table 4. Regression results for models 1 to 4.

| Dependent variable | Average master grade | | First year average master grade | |
|---|---|---|---|---|
| Model | [1] | [2] | [3] | [4] |
| Master specific | | 0.453** | | 0.415* |
| Research skills | | -1.611*** | | -1.903*** |
| Prior education | | 0.304 | | 0.273 |
| Societal impact | | -0.684** | | -0.690** |
| Interest to learn | | 0.587** | | 0.733** |
| City and University | | -0.884*** | | -0.905*** |
| Extracurricular | | -0.232 | | 0.354 |
| Word count application essay | 0.0003*** | 0.0003*** | 0.0003*** | 0.0003** |
| Average bachelor grade | 0.331*** | 0.347*** | 0.367*** | 0.388*** |
| Sex (1 = male, 2 = female) | 0.066** | 0.061** | 0.051 | 0.044 |
| Age | -0.0004 | 0.004 | -0.006 | -0.001 |
| International background: Africa | -0.678*** | -0.535*** | -0.749*** | -0.593*** |
| International background: Asia | -0.590*** | -0.378*** | -0.619*** | -0.380*** |
| International background: Central America | -0.853*** | -0.793*** | -0.924*** | -0.863*** |
| International background: Middle-East | -0.229 | -0.139 | -0.121 | -0.020 |
| International background: North America | -0.284*** | -0.161 | -0.240** | -0.103 |
| International background: South America | -0.288*** | -0.184* | -0.337*** | -0.225** |
| Master program | Yes | Yes | Yes | Yes |
| Constant | 5.179*** | 4.946*** | 4.791*** | 4.526*** |
| Observations | 823 | 823 | 820 | 820 |
| Adjusted $R^2$ | 0.289 | 0.312 | 0.296 | 0.321 |
| Residual Std. Error | 0.416 (df = 802) | 0.409 (df = 795) | 0.442 (df = 799) | 0.434 (df = 792) |
| F Statistic | 17.734*** (df = 20; 802) | 14.832*** (df = 27; 795) | 18.224*** (df = 20; 799) | 15.315*** (df = 27; 792) |

*p
**p
***p<0.01

significant amount of explained variance to the models. An inspection of the model residuals revealed that these were normally distributed. We also made a scatterplot of the residual and predicted values of the all four models to evaluate them for heteroscedasticity. Based on these plots, there was no reason to assume that heteroscedasticity occurred. We used variance inflation factors (VIF) to check for multicollinearity. All VIFs were below 2.1, indicating no problematic levels of multicollinearity.

Model 1 and 2 have the average master grade as dependent variable and models 3 and 4 have first year master grade as dependent variable. Model 1 and 3 only include the control variables, whereas models 2 and 4 also add the independent variables. When looking at model 2 and model 4 we see several significant estimates.

First, the move 'master specific' has a significant positive effect on the average grade and first year grade. This means that students that either describe mastering these contents during their bachelor in specific terms, or students that specifically state their motivation to study this content during the master acquire higher grades compared to students that write less about this. The first interpretation of this finding reinforces the notion that students with existing knowledge about the contents of their program perform better [66, 82, 83]. The second interpretation reinforces the notion that intrinsic motivation is a strong driver of study success [35] as discussed in our theoretical chapter.

The move 'research skills' has a significant negative effect on the average grade and first year grade. This negative effect is possibly explained the fact that research skills described tend to be fairly generic, which are needed for any master. This is in line with literature that shows that specific skills are more often associated with study success [84]. Then again, these generic research skills are useful for the chosen master, which makes the negative effect somewhat surprising. Another possible explanation could be that these students overestimate their own research skills, as can sometimes be the case with self-reported skills [3, 85].

The move 'prior education' has no significant effect on both average master grade and first year master grade. The move 'prior education' relates to the students qualifications, just like 'Master Specific'. However, 'prior education' is less specific about the contents and specific master program of choice, which is a likely reason that this move does not have a positive effect on study success. Further, prior education is also captured by the selection on bachelor grade, and the fact that students from only sufficiently related bachelor programs are admitted.

The move 'societal Impact' has a significant negative effect on the average grade and first year grade. Students using this move want to use knowledge gained in their master to make societal impact, which can be classified as a source of extrinsic motivation. For them, learning knowledge seems to be a means to an end, a separable outcome [35, 41] but not a goal in itself. An explanation for the negative effect has to do with our measure for study success. Grades are very good are measuring the acquisition and mastering of cognitive skills. However, making societal impact also requires some non-cognitive skills. The assessment methods of these skills are currently underdeveloped [86], which explain why there are not accounted for in the grading of the master program.

Then, the move 'interest to Learn' also has a significant positive effect on the average grade and first year grade. This effect can also be explained using motivation theory. After all, 'interest to learn' relates to the students' intrinsic motivation to study the program. They consider learning contents of the study an inherently satisfying activity, without having a separable goal in mind.

The move 'city and university' has a significant negative effect on the average grade and first year grade. The motivation to study in a particular city is an extrinsic form of motivation because students apply to enjoy the city's' university lifestyle and social networks, rather than having a strong intrinsic reason for applying to their specific program. Furthermore,

enthusiasm about the city and university is not a description of motivation for the specific master of choice. After all, there many possible masters to apply to in the city of Utrecht and it's university.

The move 'extracurricular' has no significant effect on the average grade and first year grade. We can explain this in three ways. First, 'extracurricular' is also not closely related to the specific master program, which is in line with our notion that intrinsic motivation for the specific is a determinant of study success. Second, it also does not describe the students (extrinsic) motivation for following a master in their near future, because extracurricular activities describe the students' past. Third, the skills needed to thrive in extracurricular activities might very well not be needed to thrive in an academic master program.

Looking at the control variables (Table 4, lower part), we see that the word count of the essay has a positive effect on both average and first year master grade, which corresponds with the findings of Ding (2007). The average bachelor grade also has a positive significant effect on both dependent variables. This means that students with a higher bachelor grade graduate faster, and with higher master grades. This is in accordance with existing literature on the predictive validity of grades [51, 52]. Additionally, female sex has a strongly positive effect on both average and year 1 master grade. Again, this is well in accordance with existing literature [87], likely because female students are more conscientious [88, 89]. Age of the student has no significant effect on average master grade or first year master grade. Finally, our results on the international background demonstrate that students from Africa, Asia, Central America and South America attain lower grades than students from Europe in all four models. This result corresponds with existing literature, which shows that international students, especially those from different continents can suffer from cultural bias, negatively impact their study success [90]. We also report that students from North America have significantly lower grades than European students, but only in models 1 and 3. This means this significant effect disappears when we add the independent variables to the model.

## 5. Conclusion and discussion

The aim of this paper was to answer the following research question: *What is the effect of discussing different topics in students' application essays on the student's success in a master program*? We used a Latent Dirichlet Allocation (LDA) to distill topics from students applying to a master. We distilled 30 topics from the corpus, which consists of 2701 data entries. These 30 topics were aggregated into 7 moves. We first predict the probability of each move occurring in the essay using control variables, and then we use the move probability as independent variables in our regression models. We found that five moves have a significant effect on student success when measured by grades. Students that write more about their interest to learn, and about the specific master of their choice attain higher grades. Students that write more often about their research skills, desire to make societal impact and the city and university where they intend to study attain worse grades. Writing about prior education and extracurricular activities has no significant effect on attained grades.

### 5.1. Theoretical implications

This paper provides valuable new knowledge to the student admission literature. First, we inductively study the topics and moves that are discussed in students' application essays using LDA. There are have been other studies to list themes that occur in students' application essays [26, 28], but these have been qualitative in nature. The moves we find using are text mining approach are similar to the themes discussed in existing work on applications essays (see Fig 3), and therefore provide quantitative validation of existing literature. Furthermore, existing

papers on the themes occurring in application essays list four or five themes. We find seven distinct moves, which makes our list of moves more detailed. We therefore suggest future scholars who study the contents of application essays to use our seven moves.

Furthermore, we study relationships between these moves and study success. To the best of our knowledge, we are the first to do so. Five of our distilled seven moves have a significant effect on grades attained in the master. We find that students who write about the specific contents of their master of choice and their interest to learn attain higher grades. This confirms the notion that students who demonstrate a strong intrinsic motivation to learn the contents of the program in their application essay are more successful. We report that students who write about their research skills, a more generic skill that is useful in many different masters, attain lower grades. We also report that students who write about their desire to make societal impact and the city and institute they want to study attain lower grades, since this shows they are more extrinsically motivated. For them, making societal impact or immersing themselves in the academic life is the desired outcome of following a master program rather than obtaining high grades. They have less intrinsic motivation to learn master specific knowledge and therefore attain lower grades than students who write a lot about 'Master Specific' and 'Interest to Learn'. This confirms existing theory on motivation which posits that intrinsic motivation is most effective driver of success. Second, our results lead to the conclusion that students writing about specific content and knowledge related to their master perform better than students writing about generic knowledge and skills. This finding fits well in existing literature. which shows that students with existing specific knowledge perform better [66, 91, 92].

## 5.2. Limitations

This study has three main limitations. First, the labeling of the initial thirty topics and the aggregation of these topics in moves was done manually by the three co-authors. While we used triangulation during the process, some subjectivity always remains. We encourage other authors to do replication studies, using a similar methodology, to further validate the moves occurring in application essays. The fact that our moves match well with the themes found in prior qualitative studies gives further confidence that our labeling process was robust.

Second, we did not have the application essay for every student that studied at the faculty during the period for which data collection was done. We therefore examined the composition of the groups of students with and without application essays and found no differences. This means that no type of student was over- or underrepresented in either group. Therefore, we are assured that this limitation had little effect on the outcomes of the study. Further, like most studies on student admission, we encountered admission bias. However, our data for rejected students shows that there is hardly any correlation between the probability to use any of the relevant moves in the application essay, and the average bachelor grade (Pearsons r = -0.06, n = 348), and the suitability of the bachelor program (Pearsons r = -0.08, n = 375). These correlations hardly differ from those of admitted students (average bachelor grade: Pearson's r = -0.04, n = 1582 and suitability Pearsons r = -0.02, n = 2755). In a univariate binary logistic regression, we did find a positive significant relationship between the probability of using relevant moves and being admitted or not (odds ratio = 48.99). This result is theoretically expected as the use of the relevant moves is expected to contribute to study success. Based on these results, we see little indication of admission bias.

Finally, we collected data at a faculty which spans a wide spectrum of disciplines from the social and natural sciences. In total, we studies eleven distinct master programs. However, these programs are organized by this single faculty from a single University. This brings to mind the question of generalizability. However, given the multidisciplinary nature of said

faculty we argue that our results are likely generalizable to other disciplines, faculties and institutes. Furthermore, we controlled for these disciplines with education program dummies, and found they do not significantly influence our main findings. However, further research is needed to confirm this suggestion.

## 5.3. Practical implications

We end this paper with some advice for admission committees on how to use application essays. First, we encourage committees not to dismiss application essays as admission instrument, as they are an relatively efficient tool, and because we find significant effects of several moves on student success. We argue that these moves should be carefully monitored while making the admission decision. Our most important recommendation concerns our finding that the moves we labeled 'master specific' and 'interest to learn' are positively related to study success. Based on this, we conclude that having an intrinsic motivation for the specific contents of the program is an important predictor of success. Second, the fact that 'master specific' is positively related to study success shows that discussing specific knowledge related to the program the student is applying to, as opposed to generic knowledge, is a predictor of study success. We suggest that application essays be used to evaluate both intrinsic motivation and specific knowledge. We also recommend to keep the essay in a relatively free format, as requiring the students to include specific moves will lead to a loss of discriminatory value of essays as admission instrument.

Furthermore, our study also shows that other moves that occur in application essays have no ('extracurricular' and 'prior education', or a negative ('research skills, 'societal impact' and 'city and university)' significant effect on study success. We recommend that these moves should not be given much weight during the admission decision. Finally, the negative impact of societal impact on grades could be explained by the lack of well-developed assessment methods to grade this aspect of the program. Given the societal debate for universities to become societally relevant, we urge universities to develop more assessment methods for societal impact.

## Supporting information

**S1 File. Additional regressions.**
(DOCX)

## Author Contributions

**Conceptualization:** Timon de Boer, Frank Van Rijnsoever, Hans de Bresser.

**Formal analysis:** Timon de Boer, Frank Van Rijnsoever, Hans de Bresser.

**Investigation:** Timon de Boer, Frank Van Rijnsoever.

**Methodology:** Timon de Boer, Frank Van Rijnsoever.

**Project administration:** Timon de Boer, Frank Van Rijnsoever.

**Supervision:** Frank Van Rijnsoever, Hans de Bresser.

**Validation:** Timon de Boer, Frank Van Rijnsoever.

**Visualization:** Timon de Boer.

**Writing – original draft:** Timon de Boer.

**Writing – review & editing:** Frank Van Rijnsoever, Hans de Bresser.

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
