## [Author Response · Author response to Decision Letter 0]

23 Nov 2023

Dear editor, dear reviewers,

We are grateful for the positive reception of the paper by both reviewers and the editors. Please find below how we addressed each comment. 

Comments from Editor

You should give special emphasis on the references as more recent research should be given emphasis. 

We put extra emphasis on the more recent references by changing the order of certain sentences in the introduction. Furthermore, we expanded our description of the most relevant quantitative and recent papers which our paper builds on. We hope this satisfies the editors expectations. 

Methods need to be written in more detailed as which Masters' programs were included and why? 

We have now added a description of the faculty, a the motivation for the programs that we included, and the names of the programs.

Format of the tables need to be improved.

We updated to formatting of the tables to comply with the journals guidelines.

Comments from Reviewer 1: 

1.Looking at the content, including the analysis and conclusions of the study, I suggest the re-consideration of the study’s title or using the word "what are" in place of "how" in the study title since the study deployed quantitative analysis techniques to look at the topics in application essays and their correlations and prediction of student success. ( Additional reference to Line 558; the study’s aim)

We agree with reviewer that how is not the best word in the title. Hence, we went for: “Dear admission committee…: Which moves in application essays predict student master grades?” We also opted to use the word ‘moves’ instead of ‘topics’ in the title, since this better reflects the contents of the paper 

2.Please consider reading the manuscript for some perceived minor grammatical mistakes. Example: The use of the word "motivate" in line 400 seems not to be appropriate. (actie Frank)

We understand how this can lead to confusion. We changed the sentence to: “After all, students write about master specific term to motivate how passionate they are for content of the master.” Further, we read through the manuscript and made several other minor adjustments, and spelling corrections (see manuscript.) 

3.I suggest syntax interruptions be minimised throughout the paper for easy comprehension. For instance, the sentence "Furthermore, every international background, except "international background: Africa," has a more significant positive relationship compared with our reference category, Europe. (Line 437)" could be transformed into "Furthermore, every international background has a more significant positive relationship compared with our reference category, Europe, except "Africa."

We followed this suggestion, and changed this on all occasions. 

4. Students with a relatively low bachelor’s grade and unsuitable bachelor’s degree “would probably write little about their intrinsic motivation”. (Line 616-627) Any literature to back this strong claim?

This is a very specific claim that was not found in the scarce literature on motivation letters. In the end, we decided to let the data speak. We compared the data that we have about rejected students (which we could not include in our analysis, because we have no master grades for this group). Based on this, we found no indications for admission bias:

“However, our data for rejected students shows that there is hardly any correlation between the probability to use any of the relevant moves in the application essay, and the average bachelor grade (Pearsons r = -0.06, n=348), and the suitability of the bachelor program (Pearsons r = -0.08, n=375). These correlations hardly differ from those of admitted students (average bachelor grade: Pearson’s r = -0.04, n=1582 and suitability Pearsons r = -0.02, n=2755). In a univariate binary logistic regression, we did find a positive significant relationship between the probability of using relevant moves and being admitted or not (odds ratio=48.99). This result is theoretically expected as the use of the relevant moves is expected to contribute to study success. Based on these results, we see little indication of admission bias.” 

5.For transparency purposes, any analysis carried and discussed or mentioned in the paper should be made available for readers (as a supporting document). example: "We controlled for these disciplines with our education programme."dummies, and found they do not significantly influence our main findings. "(actie Timon)

This is very good point by the reviewer. We followed this suggestion and made this analyses available in a supporting document. We did this for both the betaregressions and the OLS. 

6.We are told continents were used because the sample size was small for some regional and sub-regional groups. It is appropriate for us to know the total number of participants from each continent and whether all were adequate for the analysis carried. I suggest a table for the characteristics of the participants used, including sex, continent, age, type of master’s programme, and others available. Such information might be relevant for making appropriate inferences from the study, e.g., the deviations of African students from some of the study findings.

We included a table with means and standard deviations for the numerical variables, and distributions and standard deviations for the categorical variables as well as a correlation matrix for all numeric variables in the methodology. 

7. I doubt if the tables meet the PlosOne specifications. 

We updated to formatting of the tables to comply with the journals guidelines.

---

## [Decision Letter · Decision Letter 1]

2 Jan 2024

PONE-D-23-08316R1Dear admission committee…:

Which moves in application essays predict student master grades?PLOS ONE

Dear Dr. de Boer,

Thank you for submitting your revised manuscript to PLOS ONE. After careful consideration, we feel that it has merit but does not fully meet PLOS ONE’s publication criteria as it currently stands. Therefore, we invite you to submit a revised version of the manuscript that addresses the points raised during the review process.

We look forward to receiving your revised manuscript.

Kind regards,

Yasir Ahmad

Academic Editor

PLOS ONE

Journal Requirements:

Additional Editor Comments:

There are minor issues which if addressed will certainly improve the article and make it more suitable for the larger audience. We appreciate the work you have put in for preparation of this manuscript.

Reviewers' comments:

Reviewer's Responses to Questions

**Comments to the Author**

1. If the authors have adequately addressed your comments raised in a previous round of review and you feel that this manuscript is now acceptable for publication, you may indicate that here to bypass the “Comments to the Author” section, enter your conflict of interest statement in the “Confidential to Editor” section, and submit your "Accept" recommendation.

Reviewer #1: (No Response)

2. Is the manuscript technically sound, and do the data support the conclusions?

Reviewer #1: Yes

3. Has the statistical analysis been performed appropriately and rigorously? 

Reviewer #1: Yes

4. Have the authors made all data underlying the findings in their manuscript fully available?

Reviewer: No

5. Is the manuscript presented in an intelligible fashion and written in standard English?

Reviewer: No

6. Review Comments to the Author

Reviewer #1: While the work has improved, these are what were observed and recommended.

1.It is recommended that authors explicitly state the study design under Section 3 (Methods).

2.Considering gender as binary is becoming outdated and unacceptable. Gender, as a social construct, is not binary (male or female). Perhaps sex, a biological construct often measured as binary, could be used in place of gender in this study.

3.Please review the sentence “We connected... master” (lines 422–424) and similar ones. Conduct a thorough proofread of the entire work to ensure that all sentences are complete, easy to comprehend, and in the appropriate tense.

7. PLOS authors have the option to publish the peer review history of their article (what does this mean?). If published, this will include your full peer review and any attached files.

Reviewer: No

---

## [Decision Letter · Decision Letter 0]

7 Aug 2023

PONE-D-23-08316

Dear admission committee…

How the topics in application essays predict student master grades

PLOS ONE

Dear Dr. de Boer,

Thank you for submitting your manuscript to PLOS ONE. After careful consideration, we feel that it has merit but does not fully meet PLOS ONE’s publication criteria as it currently stands. Therefore, we invite you to submit a revised version of the manuscript that addresses the points raised during the review process.

You should give special emphasis on the references as more recent research should be given emphasis.

Methods need to be written in more detailed as which Masters' programs were included and why?

Format of the tables need to be improved.

We look forward to receiving your revised manuscript.

Kind regards,

Yasir Ahmad

Academic Editor

PLOS ONE

Journal Requirements:

Reviewers' comments:

Reviewer's Responses to Questions

Comments to the Author

1. Is the manuscript technically sound, and do the data support the conclusions?

Reviewer #1: Yes

Reviewer #2: Yes

2. Has the statistical analysis been performed appropriately and rigorously? 

Reviewer #1: Yes

Reviewer #2: Yes

3. Have the authors made all data underlying the findings in their manuscript fully available?

Reviewer #1: No

Reviewer #2: Yes

4. Is the manuscript presented in an intelligible fashion and written in standard English?

Reviewer #1: No

Reviewer #2: Yes

5. Review Comments to the Author

Reviewer #1: I recognised the efforts of the authors despite the limitations of studies of this nature. Below are my obsevations and suggestions:

1.Looking at the content, including the analysis and conclusions of the study, I suggest the re-consideration of the study’s title or using the word "what are" in place of "how" in the study title since the study deployed quantitative analysis techniques to look at the topics in application essays and their correlations and prediction of student success. ( Additional reference to Line 558; the study’s aim)

2.Please consider reading the manuscript for some perceived minor grammatical mistakes. Example: The use of the word "motivate" in line 400 seems not to be appropriate.

3.I suggest syntax interruptions be minimised throughout the paper for easy comprehension. For instance, the sentence "Furthermore, every international background, except "international background: Africa," has a more significant positive relationship compared with our reference category, Europe. (Line 437)" could be transformed into "Furthermore, every international background has a more significant positive relationship compared with our reference category, Europe, except "Africa."

4. Students with a relatively low bachelor’s grade and unsuitable bachelor’s degree “would probably write little about their intrinsic motivation”. (Line 616-627) Any literature to back this strong claim?

5.For transparency purposes, any analysis carried and discussed or mentioned in the paper should be made available for readers (as a supporting document). example: "We controlled for these disciplines with our education programme."dummies, and found they do not significantly influence our main findings. "

6.We are told continents were used because the sample size was small for some regional and sub-regional groups. It is appropriate for us to know the total number of participants from each continent and whether all were adequate for the analysis carried. I suggest a table for the characteristics of the participants used, including sex, continent, age, type of master’s programme, and others available. Such information might be relevant for making appropriate inferences from the study, e.g., the deviations of African students from some of the study findings.

7. I doubt if the tables meet the PlosOne specifications.

Reviewer #2: The manuscript section is well written and presented.

Commendable presentation of result

Good presentation of the discussion.

6. PLOS authors have the option to publish the peer review history of their article (what does this mean?). If published, this will include your full peer review and any attached files.

Do you want your identity to be public for this peer review?

 For information about this choice, including consent withdrawal, please see our Privacy Policy.

Reviewer #1: No

Reviewer #2: No

---

## [Author Response · Author response to Decision Letter 1]

26 Mar 2024

Dear editor, dear reviewers,

We are grateful for the positive reception of the paper by both reviewers and the editors. Please find below how we addressed each comment. 

Reviewer #1: 

1. It is recommended that authors explicitly state the study design under Section 3 (Methods).

We would like to thank reviewer 1 for this useful suggestion. We added the study design in the beginning of Section 3, in line 247.

2. Considering gender as binary is becoming outdated and unacceptable. Gender, as a social construct, is not binary (male or female). Perhaps sex, a biological construct often measured as binary, could be used in place of gender in this study.

This is a very good point by reviewer 1. We changes all references to ‘gender’ in the main text to ‘sex’, and added an explanatory footnote on page 16.

3. Please review the sentence “We connected... master” (lines 422–424) and similar ones. Conduct a thorough proofread of the entire work to ensure that all sentences are complete, easy to comprehend, and in the appropriate tense.

We carefully revised Section 4 of the manuscript and corrected any incorrect tenses. We now use the past tense when referring to activities that occurred in the past, and the present tense when making factual statements. We conducted a proofread of the manuscript, and enlisted a professional editor to do the same in an earlier stage. Based on this, we are confident that the work is easy to comprehend.

---

## [Editor Report · Decision Letter 2]

13 May 2024

Dear admission committee…:

Which moves in application essays predict student master grades?

PONE-D-23-08316R2

Dear Dr. de Boer,

We’re pleased to inform you that your manuscript has been judged scientifically suitable for publication and will be formally accepted for publication once it meets all outstanding technical requirements.

Kind regards,

Yasir Ahmad

Academic Editor

PLOS ONE
---

## [Editor Report · Acceptance letter]

19 Jun 2024

PONE-D-23-08316R2 

PLOS ONE

Dear Dr. de Boer, 

I'm pleased to inform you that your manuscript has been deemed suitable for publication in PLOS ONE. Congratulations! Your manuscript is now being handed over to our production team.

Kind regards, 

on behalf of

Dr. Yasir Ahmad 

Academic Editor

PLOS ONE